# Effect of Slide Diamond Burnishing on the Surface Layer of Valve Stems and the Durability of the Stem-Graphite Seal Friction Pair

**Mieczyslaw Korzynski [1], Kazimiera Dudek [1] and Katarzyna Korzynska [2,*]**

1   Centre for Innovative Technologies, Institute of Materials Engineering, College of Natural Sciences, Rzeszow University, 35-310 Rzeszow, Poland; kdudek@ur.edu.pl (K.D.)
2   Mechanical Engineering and Aeronautics Faculty, Rzeszow University of Technology, 35-959 Rzeszow, Poland
*   Correspondence: kk@prz.edu.pl

**Abstract:** This study analysed the condition of the surface layer of valve stems made of 317Ti steel after polishing and burnishing. Surface roughness, microhardness, and residual stress tests were carried out. The tests were carried out to determine the effect of the condition of the surface layer (especially non-standard parameters of surface roughness) of the stems on the durability of valves and to determine the possibility of obtaining a favourable state by means of sliding burnishing. Significant differences were observed in the values of the roughness parameters that determine the tribological properties of the surface, and higher surface microhardness and residual compressive stresses were obtained after burnishing. The durability of the stem-graphite seal in a reciprocating movement was tested, and the failure-free operation time of valves with burnished stems was approximately four times longer, which is the premise for recommending sliding diamond burnishing as a finishing treatment for valve stems.

**Keywords:** valve stem; slide diamond burnishing; surface layer; durability

## 1. Introduction

The valves of the design shown in Figure 1 have a wide range of applications for media flow control in industrial automation systems. To ensure the correct tightness of such a valve, it is important that the valve spindle, operating under sliding friction conditions in a reciprocating motion, has an optimal shape and an appropriate surface condition.

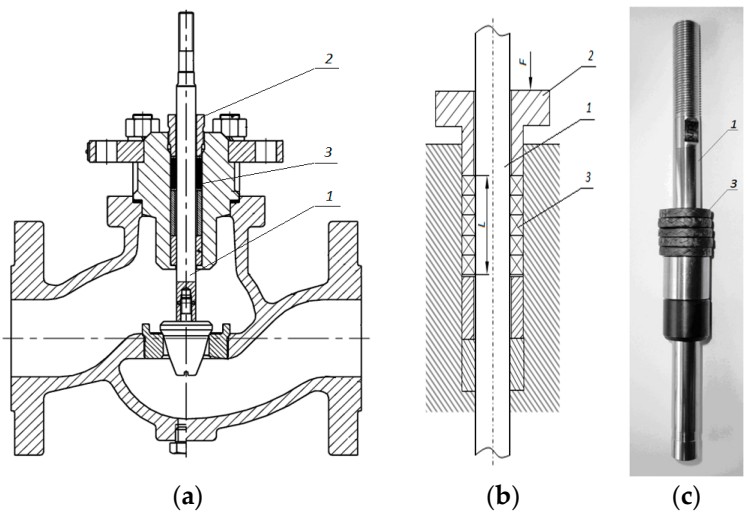

**Figure 1.** Control valve: (**a**) design overview and (**b**) valve steam-to-stuffing sealing principle; (**c**) valve stem overview: 1—valve stem, 2—gland, and 3—sealing pack.

During the initial stage of operation, the mandrel is lapped, and its working surfaces are subject to relatively quick wear. After the running-in stage, they reach the equilibrium surface texture, followed by a normal operating period, in which the wear of the friction pair becomes less intense. For a specific pair of materials and operating conditions, the amount of wear depends on the hardness and state of stresses in the surface layer and on the initial condition of the surfaces of the mating parts [1,2]. The closer the input condition of the surface layer is to an equilibrium surface texture, the less time it takes to run it in, and the longer the normal operating period. This was demonstrated in ref. [3], where the running-in time was reduced by shaping (at the finishing stage of the mandrel) the optimal surface condition by burnishing, thus extending the valve's normal operation time. In addition, in ref. [4], it was found that the condition of the mating surfaces (and the amount of wear during running-in) has a significant impact on later wear (operational), which is related to parameters $Sku$, $Sdq$, and $Svk$. The use of $Rk$ parameters to assess the surface condition to ensure lower wear and better durability of the pin-seal assembly was reported in refs. [5,6]. In ref. [7], the use of the Abbott-Firestone curve for the same purpose for hydrocylinder liners is mentioned. However, in ref. [8], as many as 17 parameters of the surface texture were used to assess tightness and wear in a reciprocating motion. This is in line with the principle that when assessing the surface condition in the context of predicting tribological properties, not only standard (commonly used) $Sa$ ($Ra$) or $Sz$ ($Rz$) parameters should be considered, but a number of other parameters that better allow the assessment of these properties should also be considered. This was confirmed in ref. [9], the results of which showed that (morphological) surface parameters such as skewness and kurtosis show a better correlation with tool wear and life than amplitude parameters such as $Sa$ and $Sq$. However, in ref. [10], parameters $Rvk$ and $Rpk$ were used to assess the degree of grinding wheel wear, and in refs. [11,12], parameters $Rvk$, $Rpk$, $Mr2$, and $Rk$ were used to assess the amount of wear during sliding friction. Investigating the relationship between the friction coefficient and some non-standard surface texture parameters, the authors of ref. [13] found that parameters $Sq$, $Ssc$, and $Sdq$ are positively correlated with the value of the sliding friction coefficient, whereas $Sal$ and $Sk$ are negatively correlated with the value of the sliding friction coefficient. The same authors in ref. [14] used a number of non-standard roughness parameters to analyse the surface texture and its resistance to micropitting. They found that the severity of micropitting was positively correlated with parameters $Sbi$, $Sku$, $Sv$, and $Sal$ but negatively correlated with $Ssc$ and $Ssk$. By examining the relevant dependencies on the four-ball tester in work [15], a relationship between parameters $St$, $Ssk$, $Spk$, and $Svk$ and the amount of wear was found. On the other hand, in the work [16] it was found that $Sku$, $Ssk$, and $Svk$ parameters showed a good correlation with the tribological properties of the mating surfaces. In ref. [17], it was also shown that parameters $Ssk$ and $Sku$ could be used to predict the tribological behaviour of the surface, and that the higher $Sku$ values and more negative Ssk values led to lower friction, which allows for the appropriate design of the surface condition and the selection of an appropriate treatment method and parameter finishing. The authors of works [18,19] came to the same conclusions, additionally stating that it is also possible to predict the wear volume surfaces using several other ($Spd$, $Std$, $Sz$, $Sdq$, $Spc$, and $Sk$) roughness parameters. On the other hand, according to the authors of paper [20], a strong relationship with the amount of wear has the parameters $Vvv$, $Svk$, $Spk$, $Sk$, $Sku$, $Sk$, and $Sq$ and according to work [21], such parameters as $Vvv$ and $Vvc$.

Similarly, in ref. [22], it was noted that low values of the height parameters, negative skewness, and kurtosis greater than three are beneficial from the point of view of lubrication of the contact surfaces. The authors of work [23] arrived at a similar conclusion by stating that the values of Ssk < 0 and Sku > 0 indicate that the surface can be considered a good bearing surface. However, in ref. [24], an increase in the values of the $Sq$, $Ssk$, and $Sku$ parameters was found to increase wear and the occurrence of scuffing.

The data presented in refs. [25–28] indicate that in the case of friction couples, the importance of the texture of the cooperating surfaces is increasingly appreciated, and

methods of its shaping are being developed. Additionally, the above-mentioned studies on sliding friction confirm the impact of non-standard texture parameters on the wear of mating surfaces. In most cases, the parameters *Ssk* and *Sku* (slope and kurtosis) are considered relevant. However, several others are taken into account depending on the type of friction, co-operating materials, lubrication conditions, etc., i.e., on the specifics (construction and working conditions) of a particular friction node, and here, opinions about the impact of individual parameters are varied. These tests were generally carried out for metal-metal friction pairs operating under the condition of unidirectional sliding friction. For the 317Ti steel-graphite gland unit of control valves operating in a reciprocating motion, no such studies have been conducted to date. In an attempt to fill this gap, studies are described in this paper to determine which surface layer properties (especially roughness parameters) are important for this friction pair and to see if they can be obtained by sliding burnishing. This research can help predict and shape the tribological properties of stems to extend the life of valves.

Grinding and polishing are the most common finishing methods used in valve stem manufacturing. They make it possible to obtain a very low surface roughness, but they leave impurities (which act as abrasive particles) on the treated surface, which causes the stems–gland pair to wear faster during operation. For small diameter stems, sometimes the above methods are replaced by turning [29], drawing, cold rolling, or ultrasonic surface rolling [30]. Drawing and cold rolling are the preferred techniques owing to their high processing output. However, they result in relatively high errors in shape, which are not favourable to the high sealing efficiency of the valves.

The research results presented in refs. [3,31,32] indicate that in the case of parts with regular shapes (particularly small-diameter shafts), slide diamond burnishing may be the preferred machining technique. The principle of burnishing the valve stem is shown in Figure 2. It is a machining method kinematically similar to turning; however, a tool with a spherical diamond tip is used here instead of a turning tool. During operation, this tool slides (with pressure) over the treated surface, causing the surface layer to cold harden and smoothing out uneven surfaces.

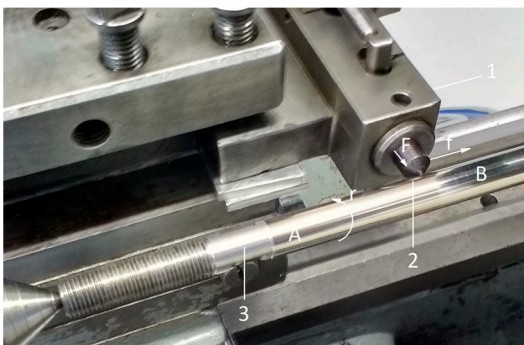

**Figure 2.** Principle of slide diamond burnishing: 1—tool holder enabling spring pressure of the tool, 2—tool (spherically ended diamond tip), 3—workpiece (valve stem), r—workpiece rotational speed, F—tool clamping force, f—tool feed, A—burnished surface, and B—previous surface.

Slide diamond burnishing enables the fabrication of hardened, extremely smooth surfaces with favourable geometrical texture characteristics, especially those characteristics that affect the tribological indices (properties) [8,22,23,33,34]. In particular, it results in smaller surface irregularities, smaller inclination angles of surface irregularities, larger radii of the rounding of irregularity peaks, and a greater load-bearing share of the surface roughness profile [3,11,30] than those obtained in the case of conventional grinding or polishing processes. These features may improve the wear resistance of treated surfaces exposed to sliding friction. This was proven in ref. [31], where an application of slide diamond burnishing for finish processing sleeve faces was reported. In this case, slide diamond burnishing resulted in a surface texture devoid of sharp irregularity peaks and

abrasive contaminants and completely covered with concentric tool marks. This ensured an exceptionally high sealing efficiency at the contact point of the sleeve faces, which could not be achieved with other standard finishing methods.

Based on the above data and considerations, it can be concluded that sliding diamond burnishing may have a similar beneficial effect on reducing the wear of friction pairs working in a reciprocating motion and thus increasing their durability. Hence, the purpose of this study was to investigate whether, as a result of the sliding burnishing treatment, it is possible to achieve a better surface condition and extend the service life of valve stems.

## 2. Test Methodology and Measurement Technique

The authors tested valve stems that were 12 mm in diameter (see Figure 1c) and made from 317Ti austenitic steel with the properties and chemical composition (EN 10088:2007 compliant) listed in Tables 1 and 2, respectively. This steel is a highly corrosion-resistant material used in valve stems and glands in various types of valves, including control valves (Figure 1a), applied in the flow control of liquids, vapours, and gases.

**Table 1.** Chemical composition (in mass %) of 317Ti steel.

| Fe | Cr | Ni | Mo | Ti | Mn | Cu | Si | V | Co | C |
|------|-------|-------|-------|-------|-------|-------|-------|-------|-------|-------|
| 66.86 | 17.13 | 10.58 | 1.903 | 0.054 | 2.016 | 0.492 | 0.459 | 0.096 | 0.077 | 0.021 |

**Table 2.** Some mechanical properties of 317Ti steel.

| Hardness, HB | Yield Limit, $R_{p0.2}$, [MPa] | Tensile Strength, $R_m$ [MPa] | Elongation $A_5$, [%] |
|--------------|-------------------------------|-------------------------------|-----------------------|
| 217 | 200 | 600 | 40 |

During operation of this valve, the stem (1) slides within the sealing part (3), the length of which is *L* (Figure 1b). Sealing of the joint between friction elements (1) and (3) is ensured by tightening the gland (2) (with a pressure of *F*, Figure 1b). With the passing of the operating time, the wear of the mating surfaces increases until a leak occurs. The faster the rate of wear, the higher the frequency of gland (2) retightening and seal replacement, and the shorter the service life of the valve. The tested seal was a pack assembly formed from stranded, square-section graphite cords woven from yarn made from expanded PTFE with incorporated graphite at a maximum share of 50% and lubricants (i.e., the seal standardly applied in industrial valves) and a valve stem. The study compared the properties of standard stems (removed from commercially available valves) with those additionally finished by diamond burnishing. It was performed on a universal turning lathe using a PCD spherical-tipped burnishing tool. Based on the recommendations given in refs. [31,35], burnishing parameters were selected to ensure surface roughness Sa < 0.2 μm, similar to that of standard stems. This surface roughness was achieved with the following processing parameters: workpiece rotational speed of 630 rpm, feed rate of 0.03 mm/r, burnishing force of 100 N, burnishing tool tip radius of 3 mm. Surface condition tests were carried out in the zone of cooperation between the stem and the gland packet (in zone 3 Figure 1c). The surface texture was tested using a Taly Scan 150 profilometer with Taly Map 2.0.15 surface analysis software, by the contact gauging method using a 5 μm profilometer tip, and the measurement results were processed in the form of contour line maps, surface photographs, 3D surface views, curves of material shares and load-bearing capacities, distributions of localised elevations, auto-correlation functions, and power spectrum density distributions. The surface layer stresses were tested using an X-ray diffractometer (X3000 Stresstech 0Y) by applying the standard method of $sin^2\psi$ [36], whereas the stress levels were calculated by applying a method of fitting with parabolic background subtraction. Axial and circumferential stress values were measured in four locations of each valve stem, and the results were expressed as the mean measurement

values. Microhardness measurements were performed in accordance with ISO 6507:2007 using the Vickers method on the Sinowon HVS-1000 device. The indenter was loaded with a force of 1 N for 15 s. The durability of the valve stems were tested by simulating normal operating conditions of a control valve with a standard gland, a flow rate of $Kvs$ = 0.63 m$^3$/h, and a valve stem displacement of h = 20 mm. These tests were performed on a test stand (Figure 3) based on a milling machine, which actuated the valve stem in a reciprocating motion. The reciprocating motion frequency was 1890 cycles/h [↓↑]. The operating medium for the control valves was hydraulic oil HL 46 (EN-ISO:6743-4) forced into the control valve by an oil pump (3) and a pressure (supply) hose (5) at 0.65 MPa. Following a period of stabilisation, which took 60 min from the start of the oil pump operation, the gland was tightened for the first time. The gland was tightened again when the joint between the valve stem and its gland was worn, which was evident from a leak of the medium. This retightening qualified as the final sealing of the control valve, and the test was continued while observing the control valve outlet and monitoring the stand using a thermal imaging camera. The criterion for durability was leakage of the medium through the outlet of the control valve. Once the leak occurred, the test was completed. The duration, number of cycles, and total displacement of the valve spindle were measured during each test.

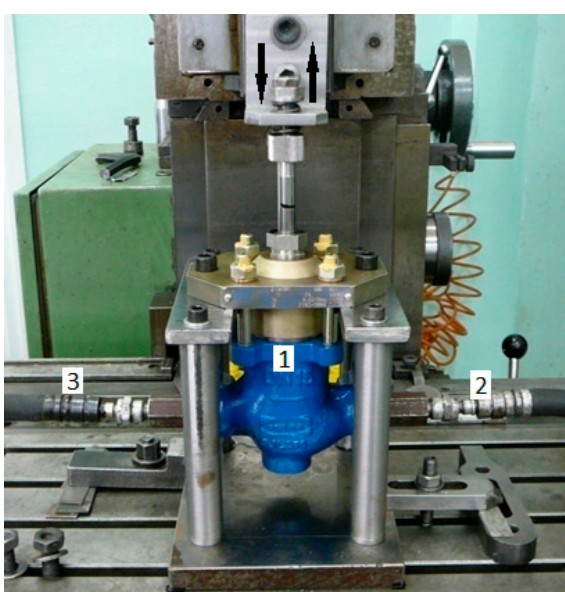

**Figure 3.** Control valve stem test rig: 1—tested control valve, 2—pressure medium supply hose for the control valve, 3—pressure medium outlet hose for the control valve, ↓↑—valve stem's direction of motion.

## 3. Results and Discussion

The test results of the surface texture (in the form of contour line maps, surface photographs, 3D surface views, curves of material shares and load-bearing capacities, distributions of localised elevations, autocorrelation functions, power spectrum density distributions, and standard and non-standard parameter values of the surfaces) of stems after both finishing methods are listed in Tables 3 and 4.

The designations of the parameters complied with EN ISO 4287:1999/A1:2010. Imaging (contour line maps, surface photography, and 3D surface views) demonstrated that the surfaces of the slide diamond-burnished specimens were more homogeneous than those of the standard stems. They also featured fewer irregularities and a lower $Sa$ value. It could be seen that the slide diamond-burnished surfaces featured symptoms of random isotropy, as confirmed by the appearance of the autocorrelation functions with a rather defined peak and the isotropy level value of $Str$ = 0.545. The density of the irregularity peaks was higher in the surfaces finished by slide diamond burnishing ($Sds$ = 2225 1/mm$^2$). It is known that, in the initial operating stage of the mating machine and equipment parts,

localised elevations of the surface have an unfavourable effect on running in. The slide diamond-burnished surfaces featured a distribution of surface irregularity peaks with a coefficient of concentration (*Sku*) of 158 and were dominated by elevations of approximately 7.5 µm, whereas the standard surfaces had a coefficient of concentration of 4.14 and were dominated by elevations of 1.2–1.4 µm. The arithmetic mean curvature of the peaks of the roughness (*Spc*) of the surface of burnished stems is about three times greater than that of standard stems. Comparing the values of roughness parameters (especially *Str*, *Sds*, and *Sku*), significant changes were found after sliding burnishing. Changes such as these are believed to have a beneficial effect on the tribological properties of frictionally cooperating machine parts [17–19,22,23].

**Table 3.** Surface geometrical structure test results for slide diamond burnished valve stems.

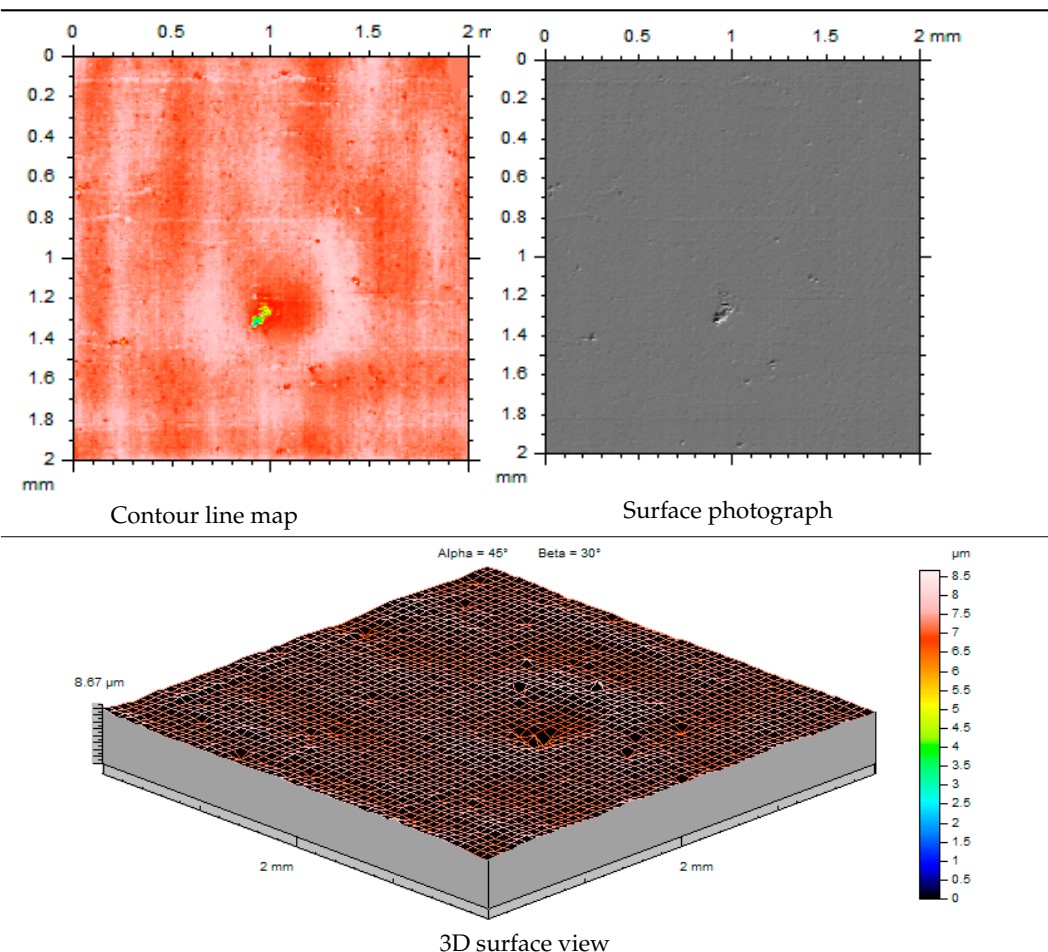

| Amplitude parameters | 3D parameters | Functional parameters |
|---|---|---|
| *Sa* = 0.12 µm | | *Sbi* = 0.193 |
| *Sq* = 0.189 µm | *Spc* = 288 1/mm | *Sci* = 1.2 |
| *Sp* = 1.23 µm | *Sds* = 2225 1/mm² | *Svi* = 0.121 |
| *Sv* = 7.44 µm | *Str* = 0.545 | *Sk* = 0.347 µm |
| *St* = 8.67 µm | *Sal* = 0.115 mm | *Spk* = 0.159 µm |
| *Ssk* = −7.17 | *Std* = 45° | *Svk* = 0.239 µm |
| *Sku* = 158 | *Sfd* = 2.12 | *Sr1* = 10.2% |
| *Sz* = 4.58 µm | | *Sr2* = 88.4% |

| Surface and volumetric parameters | Hybrid parameters |
|---|---|
| *STp* = 53.2% | *Sdq* = 0.0213 |
| *Smmr* = 0.00744 mm³/mm² | *Ssc* = 0.00444 1/µm |
| *Smvr* = 0.00123 mm³/mm² | *Sdr* = 0.0226% |

**Table 3.** *Cont.*

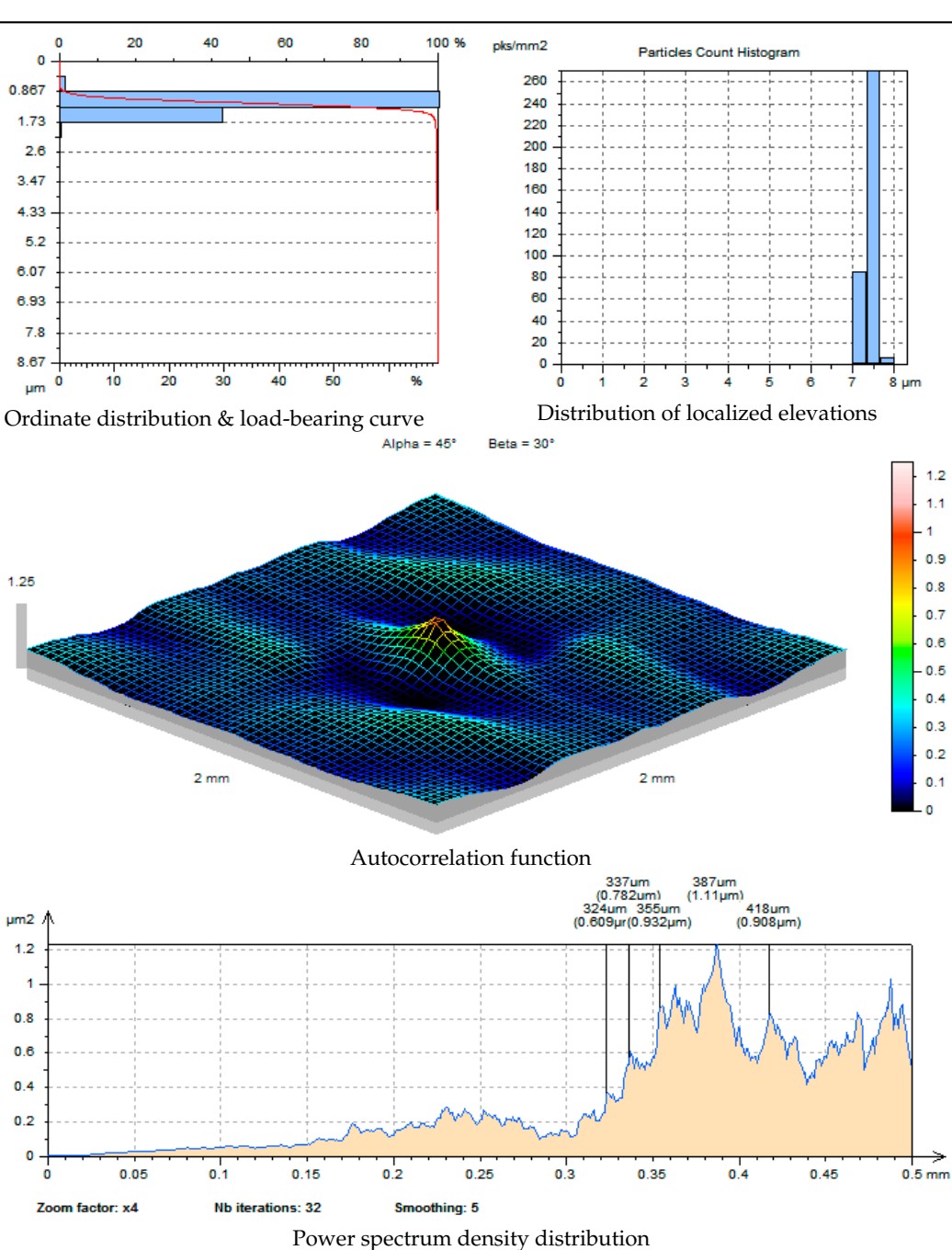

Ordinate distribution & load-bearing curve

Distribution of localized elevations

Autocorrelation function

Power spectrum density distribution

The distribution of surface ordinates in the slide diamond-burnished valve stems, which represented the changes in the material-to-void ratio as a function of height, corresponded to the surfaces with negative skewness ($Ssk = -7.17$), which were surfaces with planar-peak elevations. A comparison between the coefficients of material volume ($Smmr$) and void volume ($Smvr$) showed that the slide diamond-burnished surfaces had a higher mean material volume with a lower void volume than those of the surfaces of the standard valve stems. The cumulative distribution of the elevation ordinates (or the load-bearing curve) clearly showed a better trend for the valve stems finished by slide diamond burnishing. Their surfaces had a reduced peak height value ($Spk$), which was lower by more than 50%, indicating a potentially better abrasion resistance. Differences were also found in the core liquid retention ($Sci$), the values of which were better for slide diamond-burnished surfaces. A similar finding was made for void liquid retention ($Svi$).

Both indicated that slide diamond burnishing resulted in good levelling of the surface irregularities. Autocorrelation functions and the charts plotted for the unidimensional power spectrum density illustrate the distribution of surface irregularity deviations as a function of frequency, and (according to work [37]) suggest that the surface component values were random both in the slide diamond-burnished surfaces and the standard surfaces. An analysis of the results of surface hybrid parameter measurements (Table 5) indicated that the slide diamond-burnished valve stem surfaces had lower values of *Ssc* and *Sdq*, which indicates their larger rounding radii and lower inclination slopes than those of standard stems. These surface properties resulting from slide diamond burnishing are more favourable for the mating of couples with sliding friction and lubrication.

**Table 4.** Surface texture test results for standard valve stems.

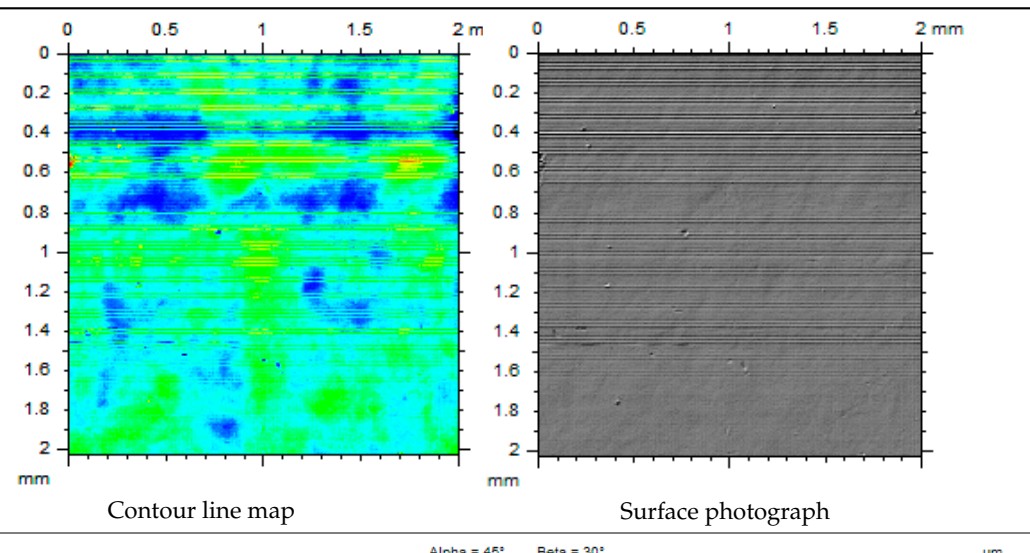

| Contour line map | Surface photograph |

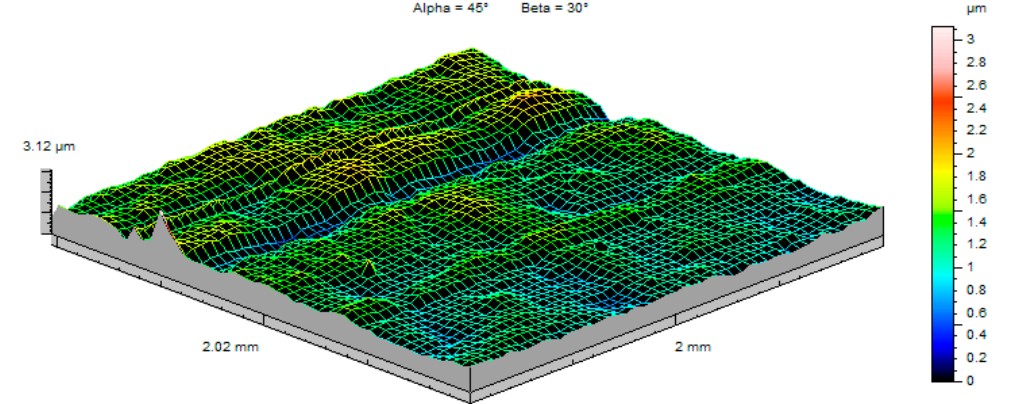

3D surface view

| Amplitude parameters | 3D parameters | Functional parameters |
| --- | --- | --- |
| *Sa* = 0.191 μm | | *Sbi* = 0.158 |
| *Sq* = 0.255 μm | *Spc* = 96.6 1/mm | *Sci* = 1.8 |
| *Sp* = 2.09 μm | *Sds* = 1799 1/mm² | *Svi* = 0.113 |
| *Sv* = 1.03 μm | *Str* = 0.203 | *Sk* = 0.506 μm |
| *St* = 3.12 μm | *Sal* = 0.0666 mm | *Spk* = 0.396 μm |
| *Ssk* = 0.486 | *Std* = 44.5° | *Svk* = 0.255 μm |
| *Sku* = 4.14 | *Sfd* = 2.53 | *Sr1* = 15.4% |
| *Sz* = 2.17 μm | | *Sr2* = 88.2% |

**Table 4.** *Cont.*

| Surface and volumetric parameters | Hybrid parameters |
|---|---|
| $STp = 46\%$<br>$Smmr = 0.00103 \text{ mm}^3/\text{mm}^2$<br>$Smvr = 0.00209 \text{ mm}^3/\text{mm}^2$ | $Sdq = 0.0566$<br>$Ssc = 0.0127 \text{ 1}/\mu\text{m}$<br>$Sdr = 0.159\%$ |

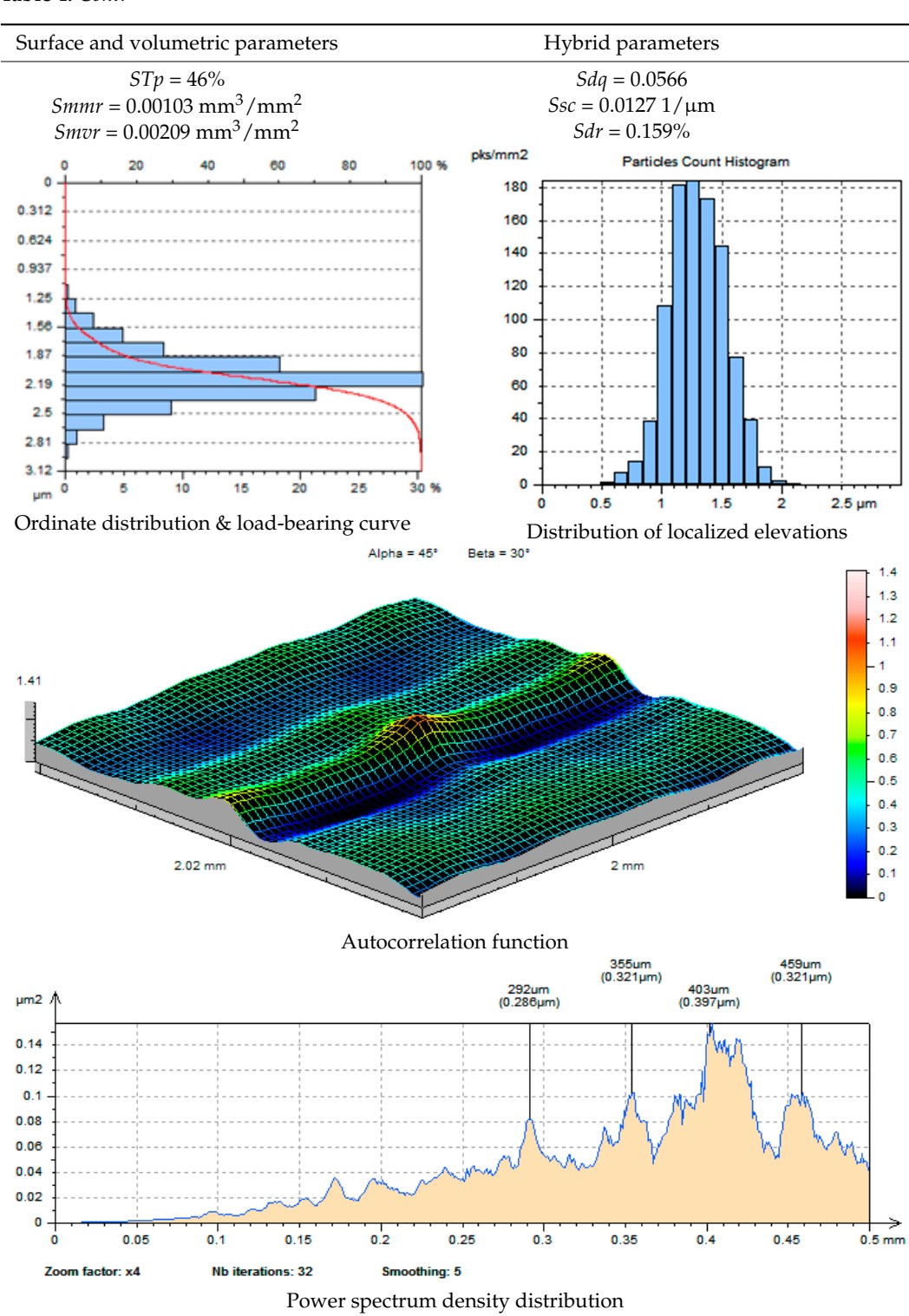

Ordinate distribution & load-bearing curve

Distribution of localized elevations

Autocorrelation function

Power spectrum density distribution

The microhardness and stress levels of the slide-diamond-burnished and standard surfaces of the valve stems are listed in Table 6. The sliding diamond burnishing operation was found to increase the superficial microhardness, which should reduce the rate of wear during operation with friction. All the tested stems were found to feature compressive stresses, which, however, were much higher (in absolute value) after slide burnishing. Compressive stresses are important features of the superficial layer and improve abrasion resistance and the mating of coupled parts that are exposed to corrosion-promoting

factors. High compressive stresses inhibit the penetration of corrosive agents through the surface [38] and may reduce the rate of corrosion processes. Hence, the stress state produced by slide diamond burnishing is deemed beneficial.

**Table 5.** Values of parameters describing irregularity.

| Valve Stem Version | Tested Parameters | | | |
|---|---|---|---|---|
| | $Ssc$ [1/µm] | $Sdq$ | $R_w$ [µm] | $\beta$ [°] |
| Standard | 0.0127 | 0.0566 | 78.74 | 3.24 |
| Slide diamond burnished | 0.00444 | 0.0213 | 225.225 | 1.22 |

**Table 6.** Stress level and microhardness test results.

| Valve Stem Version | Stress Level [MPa] | | | | Microhardness [MPa] | |
|---|---|---|---|---|---|---|
| | Circumferential | Standard Deviation | Axial | Standard Deviation | | Standard Deviation |
| Standard | −305.9 | 13.8 | −510.9 | 24.4 | 326 | 6 |
| Slide diamond burnished | −479.6 | 12.9 | −773.9 | 15.5 | 409 | 19 |

The valve stems analysed in this paper were subjected to durability tests and, in order to better assess the impact of burnishing parameters and tribological surface texture parameters, the durability of the stems burnished with slightly different conditions was also tested (Table 7). These conditions were selected so as to obtain a similar surface roughness Sa and different values of the other texture parameters. From the data presented in Table 7, it can be concluded that there is a clear convergence of the values of surface roughness parameters with the durability of the tested valve stems and that optimization would be useful in future research. The found differences in durability prove that the slide diamond burnishing of the working surface of the valve stems may produce a beneficial condition for the surface layer, in which all parameters, without exception, are conducive to the improvement of tribological properties, which can multiply the MTBF of tested friction couples.

**Table 7.** Texture parameters of tested stems and the results of durability tests of the valve stem and gland couples exposed to sliding friction.

| No. | | 1 | 2 | 3 | 4 |
|---|---|---|---|---|---|
| Valve Stem Version | | Standard | Slide Diamond Burnished | | |
| Burnishing parameters | Tool tip radius [mm] | – | 3 | 4 | 4 |
| | Feed rate [mm/r] | – | 0.03 | 0.11 | 0.03 |
| | Force [N] | – | 100 | 150 | 50 |
| Texture parameters | $Sa$ [µm] | 0.191 | 0.12 | 0.153 | 0.209 |
| | $Sz$ [µm] | 2.17 | 4.58 | 7.22 | 8.92 |
| | $Sku$ | 4.14 | 158 | 83.3 | 77 |
| | $Ssk$ | 0.486 | −7.17 | −0.604 | −6.8 |
| | $Sdq$ | 0.0566 | 0.0213 | 0.0355 | 0.0484 |
| | $Sal$ [mm] | 0.067 | 0.115 | 0.044 | 0.042 |
| | $Sbi$ | 0.158 | 0.193 | 0.091 | 0.238 |
| | $Sq$ [µm] | 0.255 | 0.189 | 0.262 | 0.435 |

**Table 7.** *Cont.*

| No. | | 1 | 2 | 3 | 4 |
|---|---|---|---|---|---|
| **Valve Stem Version** | | **Standard** | **Slide Diamond Burnished** | | |
| Texture parameters | $Sv$ [μm] | 1.03 | 7.44 | 6.19 | 8.61 |
| | $Ssc$ [1/μm] | 0.01271 | 0.00444 | 0.00561 | 0.00557 |
| | $St$ [μm] | 3.12 | 8.67 | 9.39 | 10.8 |
| | $Spk$ [μm] | 0.396 | 0.159 | 0.164 | 0.237 |
| | $Svk$ [μm] | 0.255 | 0.239 | 0.465 | 0.903 |
| Durability test results (until failure by leaking) | Valve working time [h] | 72 | 312 | 148 | 103 |
| | Friction distance [m] | 5200 | 22,530 | 10,690 | 7340 |
| | Number of cycles | 14,000 | 60,670 | 28,780 | 20,030 |

## 4. Conclusions

- The test results substantiated the conjecture that the surface layer condition produced by slide diamond burnishing has a beneficial influence to the durability of the friction couple formed by the 317Ti austenitic steel stem and graphite cord, which works with a reciprocating motion. With properly selected burnishing parameters, the durability of such valves can be up to four times longer than in the case of standard valves (with a drawn stem), which is a premise for recommending sliding diamond burnishing as a finishing treatment for valve stems.
- Compared to standard machining (drawing), diamond burnishing of valve stems resulted in surfaces with increased load capacity, with elevations characterized by about three times larger radii of rounding of the tops, and (also about three times) gentler inclination of the slopes of the tops. These changes, as well as higher values of kurtosis and significant negative values of surface skewness after burnishing, resulted in better adaptation of the stems to work in conditions of reciprocating friction.
- After sliding burnishing, approximately 25% higher surface microhardness and much higher (by approximately 60%) compressive stress than in standard stems were noted. The synergy of these effects and changes in the surface texture contributed to the increased durability of the valves.
- The research results obtained in this work are a premise for further research (e.g., optimization) on the legitimacy of using sliding diamond burnishing as finishing machining of 317Ti austenitic steel parts exposed to sliding friction.

**Author Contributions:** Conceptualization, M.K.; methodology, K.D.; analysis, K.K. and M.K.; investigation, K.D.; writing—original draft preparation, K.K. and K.D.; writing—review and editing, M.K.; visualization, K.K.; supervision, M.K.; project administration, K.K. All authors have read and agreed to the published version of the manuscript.

**Funding:** This research received no external funding.

**Institutional Review Board Statement:** Not applicable.

**Informed Consent Statement:** Not applicable.

**Data Availability Statement:** Not applicable.

**Conflicts of Interest:** The authors declare no conflict of interest.

## Nomenclature

The following abbreviations and surface topography parameters are used in the manuscript:

| | |
|---|---|
| MTBF | Mean time between failures |
| PCD | Polycrystalline diamond |
| PFTE | Polytetrafluoroethylene |
| *Sa* | Arithmetic mean height, μm |
| *Sal* | Auto-correlation length, mm |
| *Sbi* | Surface bearing index |
| *Sci* | Core fluid retention index |
| *Sdq* | Root mean square gradient |
| *Sdr* | Developed interfacial areal ratio, % |
| *Sds* | Density of summits of the surface, pks/mm$^2$ |
| *Sfd* | Fractal dimension |
| *Sk* | Core roughness depth, μm |
| *Sku* | Kurtosis |
| *Sp* | Maximum peak height, μm |
| *Spc* | Arithmetic mean peak curvature, 1/mm |
| *Spd* | Peak density, 1/mm$^2$ |
| *Spk* | Reduced summit height, μm |
| *Sq* | Root mean square height, μm |
| *Sds* | Density of summits of the surface, pks/mm$^2$ |
| *Smmr* | Average volume of elevated material in a unit area, mm$^3$/mm$^2$ |
| *Smvr* | Average volume of cavities on a unit area, mm$^3$/mm$^2$ |
| *Sr1* | Upper bearing surface, % |
| *Sr2* | Lower bearing surface, % |
| *Ssc* | Arithmetic mean summit curvature of the surface, 1/μm |
| *Ssk* | Skewness |
| *St* | Total height of roughness profile, μm |
| *STp* | Surface material ratio, % |
| *Std* | Texture direction, ° |
| *Str* | Texture parameter |
| *Sv* | Maximum valley depth, μm |
| *Svi* | Valley fluid retention index |
| *Svk* | Reduced valley depth, μm |
| *Sz* | The maximum height of surface, μm |
| *Vvv* | Dale void volume, mm$^3$/mm$^2$ |
| *Vvc* | Core void volume, mm$^3$/mm$^2$ |

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
