# Peer review of "Effect of Slide Diamond Burnishing on the Surface Layer of Valve Stems and the Durability of the Stem-Graphite Seal Friction Pair"

_applsci, doi:10.3390/app13116392_

Round 1

Reviewer 1 Report

The paper presents the effect of sliding diamond burnishing on the surface layer of valve stems and the durability of the stem-graphite seal friction pair. This paper is of great importance for industrial practice, but to be accepted several problems must be solved, namely:

1. There are several sentences and words that I consider to be reformulated, because in the natural language I think they sound good, but in English they are difficult to understand. That is why the English language must be rechecked with someone well versed in technical English.

2. When codifications (notations) of a some parameters or sizes appear for the first time, what it means/represents is written and then they can be written as notation/codification, and there are enough of them. 

Perhaps it would be simpler to introduce a list of notations at the beginning, for an understanding, by anyone!

3. In line 115, not "...(ref. Fig. 1c)..." but '... (see Fig. 1c) ...'

4. In line 118, use another word instead of "glands/gland", maybe it's correct is 'press gaskets/press gasket' respectively in Fig. 1, position 2!

This should be changed throughout the text of the paper.

5. At turn 170 Figure 4, don't say anything, make a statement about what this figure represents, not what you write now in the test

6. In line 171, why "Discussion of the test results" and not "Results and discussions"?

7. Tables 3 and 4 are a mixture of figures and some results that can be tabulated. I propose to separate the images as Figures with positions a, b, c, ... and tabulate the results!

8. On line 192, "... mating machine..." what is this machine? I don't think it's right! Must be checked!

9. In line 196, what does 270 pks/mm2 represent? Similar in Tables 3 and 4!

10. In line 199-219, make several comparisons, e.g. "Tribological parameters values such as Str, Sds, and Sku were compared (with whom?; and where?) and found to be significantly better when surfaces were finished by slide diamond burnishing."

Other comparisons follow, but it does not result with whom and where are they presented?

I also made a reference to reference [25] and it is not clear what it helped you with? You used results and compared them with your own, or....

11. In the "Conclusions" avoid using the word "peak/peaks" lines 257-258, but 'roughness' as you used in the Abstract and do not repeat it several times in the same sentence!

Then, the Abstract ends with a conclusion that no longer appears in the "Conclusions", why?

12. I think that in order to improve this paper, I recommend that you also consult the following works:

- for polishing with a single diamond by sliding on the surface layer, the work with: https://doi.org/10.3390/ma16072550

- for surface roughness parameters that determine tribological properties, the work with: DOI 10.1179/1751584X13Y.0000000044.

The English language is sometimes difficult to understand in some sentences. I have a feeling that in the natural language it sounds good, but in technical English it has a different meaning. The authors also use some technical words that give a different meaning in technical English!

Author Response

The manuscript has been rechecked and appropriate changes have been made in accordance with suggestions. The responses to comments have been prepared and attached herewith (our responses are in blue colour) and in the revised version of the paper, the changed places are highlighted in yellow. We thank you for your thoughtful suggestions and insights, which have enriched the manuscript and produced a better and more balanced account of the research. We hope that the revised manuscript is now suitable for publication in Applied Sciences.

On behalf of the authors - K. Korzynska

Rev. 1.

The paper presents the effect of sliding diamond burnishing on the surface layer of valve stems and the durability of the stem-graphite seal friction pair. This paper is of great importance for industrial practice, but to be accepted several problems must be solved, namely:

  1. There are several sentences and words that I consider to be reformulated, because in the natural language I think they sound good, but in English they are difficult to understand. That is why the English language must be rechecked with someone well versed in technical English.

We have included an explanation at the end of these notes.

  1. When codifications (notations) of a some parameters or sizes appear for the first time, what it means/represents is written and then they can be written as notation/codification, and there are enough of them. 

Perhaps it would be simpler to introduce a list of notations at the beginning, for an understanding, by anyone!

There are few of them, they are usually standard designations, used in technology and commonly known (certainly to AS readers). We hope the reviewer shares this opinion.

  1. In line 115, not "...(ref. Fig. 1c)..." but '... (see Fig. 1c) ...'

Corected.

  1. In line 118, use another word instead of "glands/gland", maybe it's correct is 'press gaskets/press gasket' respectively in Fig. 1, position 2!

This should be changed throughout the text of the paper.

The term gland is more commonly used to seal moving connections. ("Gland packing is made of a gland rope"). The gasket is rather used to seal fixed connections. So, accept the gland term please.

  1. At turn 170 Figure 4, don't say anything, make a statement about what this figure represents, not what you write now in the test.

In the current, corrected version of the paper, Fig. 4, has been removed because it is indeed of little relevance to the discussion. 

  1. In line 171, why "Discussion of the test results" and not "Results and discussions"?

Corrected according suggestion.

  1. Tables 3 and 4 are a mixture of figures and some results that can be tabulated. I propose to separate the images as Figures with positions a, b, c, ... and tabulate the results!

If the reviewer agrees, the authors suggest leaving tables 3 and 4 as this form is more concise, allows for a faster review of the results and is used in publications.

  1. On line 192, "... mating machine..." what is this machine? I don't think it's right! Must be checked!

The text says "mating machine parts". To be clear, in corrected version is written: "mating parts".

  1. In line 196, what does 270 pks/mm2 represent? Similar in Tables 3 and 4!

In the revised version of the article, these parts of text have been improved to be clearer. The relevant data has also been corrected in the text and in the tables mentioned.

  1. In line 199-219, make several comparisons, e.g. "Tribological parameters values such as Str, Sds, and Sku were compared (with whom?; and where?) and found to be significantly better when surfaces were finished by slide diamond burnishing."Other comparisons follow, but it does not result with whom and where are they presented?

In the revised version of the article, we have tried to improve these sentences to make them clearer.

I also made a reference to reference [25] and it is not clear what it helped you with? You used results and compared them with your own, or....

In that book there was a statement "surfaces with negative skewness are planar-peak elevations". Since this is self-explanatory and it does not require a reference to the literature, so this reference has been removed.

  1. In the "Conclusions" avoid using the word "peak/peaks" lines 257-258, but 'roughness' as you used in the Abstract and do not repeat it several times in the same sentence!
    Then, the Abstract ends with a conclusion that no longer appears in the "Conclusions", why?

The conclusions were edited anew, we tried to take into account the comments of the reviewers.

  1. I think that in order to improve this paper, I recommend that you also consult the following works:

- for polishing with a single diamond by sliding on the surface layer, the work with: https://doi.org/10.3390/ma16072550

  • - for surface roughness parameters that determine tribological properties, the work with: DOI 10.1179/1751584X13Y.0000000044. (https://doi.org/10.1179/1751584X13Y.0000000044)

Thank you for pointing out these interesting publications. We regret that we did not deal with the issues of selective transfer, friction energy, tribochemiac or thermal effects. The issues and research methods described there will help us in future research.

Comments on the Quality of English Language

The English language is sometimes difficult to understand in some sentences. I have a feeling that in the natural language it sounds good, but in technical English it has a different meaning. The authors also use some technical words that give a different meaning in technical English!

Our paper was translated and rechecked by ELES (see below). We sent them a request to recheck the text or send a language certificate, but so far we have not received a response. Since the deadline required by the AS editorial office is already passing, we send the article with corrections made by us. We hope it will be accepted.

Temat: Order delivery: Language Editing Standard - Order reference
ASLESTD0333657
Data: 2023-02-10 13:30
Od: [email protected]
Do: [email protected]
             Your edited document is ready
Dear Katarzyna Korzynska,
Your edited document is now ready, please save the file to your desktop.
The download link will expire after 6 months.
 Order reference: ASLESTD0333657
Language Editing Standard
                Edited Document
 Save file to desktop [2]
-------------------------
 Please note:
 If you make any alterations to your manuscript after we return it to
you and your manuscript is rejected by a journal for language errors,
you cannot return it to us for free re-editing. However if you do not
alter your manuscript that we have edited and it is rejected by a
journal on purely English Language grounds, you can receive either a
full refund or have your message re-edited free of charge.
-------------------------
 We wish you the best of luck with your article and thank you for
choosing Elsevier Language Editing Services
If you have any other questions, please contact us at
[email protected]
Kind regards
Elsevier

Reviewer 2 Report

In this paper, the effect of sliding diamond burnishing on the surface quality of valve stems is analyzed. The durability of valve stems made using both standard methods and sliding diamond burnishing were tested in experiments.

General comments

·         In the introduction section, the shortcoming of the reviewed literature is not clarified.

·         References used for obtaining the data in tables 1 and 2 should be mentioned.

Technical comments

·         Authors should clarify the novelty of the paper; they should identify the gaps in the available literature and describe how this research tries to address those gaps.

·         In section 2, the process parameters used for obtaining the lowest surface roughness are given. However, it is not clear how these parameters were obtained. Authors should present the design of experiments, including the number of tests and the parameters used in the tests, that was used for achieving these values.

·         Also in section 2, operation testing conditions for the manufactured valve stems are presented. However, it is note clear why these conditions were used. Is this the standard method for testing these valves?

·         The caption for figure 4 is not correct. Also, it is not clear how this figure, i.e., valve and oil temperature, is relevant to the discussions.

·         Regarding tables 3 and 4, authors should clarify the location on the valve stem where surface tests were performed. Also, the definitions of the surface roughness and morphology metrics presented in these tables should be offered.

·         In table 6, authors should clarify how the results were obtained. What were the testing conditions and characteristics of the equipment used for the tests?

·         In table 7, what was the basis for choosing the burnishing parameters?

·         The novelties and contributions of the research are not clear. The fact that an extra finishing process improves the surface quality is trivial. Also, optimal process conditions for manufacturing the valve stems are not obtained and standard methods for testing the stems are not followed.

In conclusion, I believe this paper is not suitable for publication in its current form.

The paper requires a thorough language review. Some language errors exist in the text that need to be fixed.

Author Response

The manuscript has been rechecked and appropriate changes have been made in accordance with suggestions. The responses to comments have been prepared and attached herewith (our responses are in blue colour) and in the revised version of the paper, the changed places are highlighted in yellow. We thank you for your thoughtful suggestions and insights, which have enriched the manuscript and produced a better and more balanced account of the research. We hope that the revised manuscript is now suitable for publication in Applied Sciences.

On behalf of the authors - K. Korzynska

Rev. 2.

In this paper, the effect of sliding diamond burnishing on the surface quality of valve stems is analyzed. The durability of valve stems made using both standard methods and sliding diamond burnishing were tested in experiments.

 General comments

  •         In the introduction section, the shortcoming of the reviewed literature is not clarified.

We tried to improve it. Hopefully now it will be accepted.

  • References used for obtaining the data in tables 1 and 2 should be mentioned.

Corrected. The relevant standard is mentioned in the first sentence of chapter 2.

 Technical comments

  • Authors should clarify the novelty of the paper; they should identify the gaps in the available literature and describe how this research tries to address those gaps.

According to this remark, chapter 1 of the work was supplemented. We hope that the supplement meets the reviewer's requirements.

  • In section 2, the process parameters used for obtaining the lowest surface roughness are given. However, it is not clear how these parameters were obtained. Authors should present the design of experiments, including the number of tests and the parameters used in the tests, that was used for achieving these values.

These parameters have been selected based on published results our previous experiences. A related note is included in the text of corrected paper.

  • Also in section 2, operation testing conditions for the manufactured valve stems are presented. However, it is note clear why these conditions were used. Is this the standard method for testing these valves?

Durability tests are not standard methods. Only surface roughness (Sa) and stem dimensions are checked at the manufacturing plant prior to valve assembly. Valves are randomly tested for tightness. It is obvious that our (burnishing) grips met all technical requirements, so we do not describe it.

  • The caption for figure 4 is not correct. Also, it is not clear how this figure, i.e., valve and oil temperature, is relevant to the discussions.

Sorry, there was a mistake in the caption for fig. 4. However, in the current, revised version of the article, figure 4 has been removed as it is irrelevant to the discussion.

  • Regarding tables 3 and 4, authors should clarify the location on the valve stem where surface tests were performed. Also, the definitions of the surface roughness and morphology metrics presented in these tables should be offered.

Surface condition tests were carried out in the place of cooperation of the pin with the sealant (in zone 3 Fig. 1c). A corresponding note has been included in the text. We would like to point out that the entire surface of the mandrel was machined equally and the place of taking measurements is not so important.

The surface roughness and morphology metrics mentioned by the reviewer appear in Tabs 3-6 and their definitions would have to be repeated there. They could be included as a separate "a list of notations", but we believe that they are widely known, especially to AS readers, and the reference to the standard in the text is sufficient. We hope the reviewer shares this view and will accept this argument.

  • In table 6, authors should clarify how the results were obtained. What were the testing conditions and characteristics of the equipment used for the tests?

The methodology is standard, it has been presented in the text in Chapter 2, more detailed data on measurements have been added in corrected version of paper.

  • In table 7, what was the basis for choosing the burnishing parameters?

The burnishing parameters were chosen based on previous trials and experiments. They did not have to give specific results, it was enough to obtain similar Sa roughness and slightly different other surface roughness parameters. This was added in the revised text.

  • The novelties and contributions of the research are not clear. The fact that an extra finishing process improves the surface quality is trivial. Also, optimal process conditions for manufacturing the valve stems are not obtained and standard methods for testing the stems are not followed.

We have tried to make these issues a little clearer with the third (added) paragraph in Chapter 1.

 In conclusion, I believe this paper is not suitable for publication in its current form.

We hope that the corrected paper will be accepted for publication in its current form.

Comments on the Quality of English Language

The paper requires a thorough language review. Some language errors exist in the text that need to be fixed.

Our article was translated and reviewed by ELES (see below). We sent them a request to recheck the text or send a language certificate, but so far we have not received a response. Since the deadline required by the AS editorial office is already passing, we send the article with corrections made by us. We hope it will be accepted.

Temat: Order delivery: Language Editing Standard - Order reference
ASLESTD0333657
Data: 2023-02-10 13:30
Od: [email protected]
Do: [email protected]
             Your edited document is ready
Dear Katarzyna Korzynska,
Your edited document is now ready, please save the file to your desktop.
The download link will expire after 6 months.
 Order reference: ASLESTD0333657
Language Editing Standard
                Edited Document
 Save file to desktop [2]
-------------------------
 Please note:
 If you make any alterations to your manuscript after we return it to
you and your manuscript is rejected by a journal for language errors,
you cannot return it to us for free re-editing. However if you do not
alter your manuscript that we have edited and it is rejected by a
journal on purely English Language grounds, you can receive either a
full refund or have your message re-edited free of charge.
-------------------------
 We wish you the best of luck with your article and thank you for
choosing Elsevier Language Editing Services
If you have any other questions, please contact us at
[email protected]
Kind regards
Elsevier

Reviewer 3 Report

1) The English text was difficult to understand in many places. It seems that the text wasn’t proofread properly.

2) Abstract must mention the specifying the purpose.

3) The manuscript introduction text needs to be updated.  Of the 26 references used in the introductory text, only 6 were published in recent years. In addition, this is a problem that extends throughout the entire manuscript; However; Seemingly, a comprehensive literature review was not given. Compared with the existing research, what are the main research contributions of the present research? The authors have to stop after writing each example and think about the contributions and lack of knowledge for each paper. After that, in the final lines of the introduction give the blank spots of the topic. More references should be included certainly the reference.

Some key references were studied in several paper as follows:

(#1) An Investigation of Thermal Properties of Zirconia Coating on Aluminum url:

https://link.springer.com/article/10.1007/s13369-012-0289-z,

(#2) Investıgatıon Of The Wear Characterıstıcs Of Thermal Barrıer Coatıng In A Bıodıesel Engıne

url:https://www.worldscientific.com/doi/abs/10.1142/S0218625X19501580

4) The novelty of the paper may be elaborated before  Materials and methods. Rewrite the paragraph. I also suggest further discussing the relevance of the work.

5) The discussion of the scientific findings in this paper is inadequate.

6) The conclusions are formulated very general without summarizing what was important in the work.

Authors should carefully study the comments and make improvements to the article step by step. All changes should be highlighted in color.

Author Response

The manuscript has been rechecked and appropriate changes have been made in accordance with suggestions. The responses to comments have been prepared and attached herewith (our responses are in blue colour) and in the revised version of the paper, the changed places are highlighted in yellow. We thank you for your thoughtful suggestions and insights, which have enriched the manuscript and produced a better and more balanced account of the research. We hope that the revised manuscript is now suitable for publication in Applied Sciences.

On behalf of the authors - K. Korzynska

Rev. 3.

1) The English text was difficult to understand in many places. It seems that the text wasn’t proofread properly.

We include a separate explanation for language. See at the end, please.

2) Abstract must mention the specifying the purpose.

The abstract has been supplemented in accordance with this suggestion.

3) The manuscript introduction text needs to be updated.  Of the 26 references used in the introductory text, only 6 were published in recent years. In addition, this is a problem that extends throughout the entire manuscript; However; Seemingly, a comprehensive literature review was not given. Compared with the existing research, what are the main research contributions of the present research? The authors have to stop after writing each example and think about the contributions and lack of knowledge for each paper. After that, in the final lines of the introduction give the blank spots of the topic. More references should be included certainly the reference.

References have been supplemented with new items. The literature review in Chapter 1 has been expanded and summarized. We hope that this has been done in accordance with the reviewer's recommendations to the extent that he will accept the revised text.

Some key references were studied in several paper as follows:

(#1) An Investigation of Thermal Properties of Zirconia Coating on Aluminum url:

https://link.springer.com/article/10.1007/s13369-012-0289-z,

(#2) Investıgatıon Of The Wear Characterıstıcs Of Thermal Barrıer Coatıng In A Bıodıesel Engıne

          url:https://www.worldscientific.com/doi/abs/10.1142/S0218625X19501580

Thank you for pointing out these interesting publications. The issues and research methods described on coatings and thermal issues there will help us in future research.

4) The novelty of the paper may be elaborated before  Materials and methods. Rewrite the paragraph. I also suggest further discussing the relevance of the work.

In accordance with these suggestions, the supplemented text in chapters 1 and 2.

5) The discussion of the scientific findings in this paper is inadequate.

We have slightly improved the text, but the broader discussion, however, requires deeper research, which we are currently continuing and may publish in the future; we ask for your understanding on this issue.

6) The conclusions are formulated very general without summarizing what was important in the work.

In the revised version of the paper, we have tried to improve the conclusions hoping that they will be more acceptable to the reviewer.

Authors should carefully study the comments and make improvements to the article step by step. All changes should be highlighted in color.

This has been done.

Regarding point 1:

Our paper was translated and rechecked by ELES (see below). We sent them a request to recheck the text or send a language certificate, but so far we have not received a response. Since the deadline required by the AS editorial office is already passing, we send the article with corrections made by us. We hope it will be accepted.

Temat: Order delivery: Language Editing Standard - Order reference
ASLESTD0333657
Data: 2023-02-10 13:30
Od: [email protected]
Do: [email protected]
             Your edited document is ready
Dear Katarzyna Korzynska,
Your edited document is now ready, please save the file to your desktop.
The download link will expire after 6 months.
 Order reference: ASLESTD0333657
Language Editing Standard
                Edited Document
 Save file to desktop [2]
-------------------------
 Please note:
 If you make any alterations to your manuscript after we return it to
you and your manuscript is rejected by a journal for language errors,
you cannot return it to us for free re-editing. However if you do not
alter your manuscript that we have edited and it is rejected by a
journal on purely English Language grounds, you can receive either a
full refund or have your message re-edited free of charge.
-------------------------
 We wish you the best of luck with your article and thank you for
choosing Elsevier Language Editing Services
If you have any other questions, please contact us at
[email protected]
Kind regards
Elsevier

Round 2

Reviewer 1 Report

 It is true that they are standard and well-known names, but it is not wrong to have presented a list with notations. It is possible to read specialists from other fields or close to the field, and then...

Then, the purpose of the work is not yet clearly defined, what is the aim of the work? What is being followed?

Instead of putting a unit of measure 1/mm2, why not put mm-2? There are other notations that have no units of measure, simply a value, like a coefficient! Are they really all dimensionless? Check again! I insist on a list of notations and units of measure!

Now you say "Comparing the measurement results of tribological parameters such as Str, Sds, and Sku, they were found to be significantly better when the surfaces were finished by diamond burnishing."; on what basis are the parameters Str, Sds, and Sku better? The comparison has no scientific and practical support! Statements are not enough, there must be a clear argumentation! This is also valid for other parameters!

I indicated some sources to find that there are concerns with polishing with a single abrasive, values of the tribological parameters, and very good tribological properties also through the Chemical-Mechanical Polishing process. You don't necessarily have to worry about the selective transfer, but polished surfaces can also be obtained through other processes!

Author Response

We thank you for your again review and thoughtful suggestions and insights, which have enriched the manuscript and produced a better and more balanced account of the research.

Below, your comments are written in black. Our responses (written in blue) have been attached herewith.

We did not mark changes in the paper, as these we explained in detail in our reply.

We hope that the revised manuscript is now suitable for publication in Applied Sciences journal.

Reviewer 2 Report

The manuscript can now be accepted for publication.

Author Response

We are grateful and thank you for your thoughtful suggestions and insights, which have enriched the manuscript and produced a better and more balanced account of the research.

Katarzyna Korzynska